

# Zonula occludens and nasal epithelial barrier integrity in allergic rhinitis

Che Othman Siti Sarah[1], Norasnieda Md Shukri[2], Noor Suryani Mohd Ashari[1] and Kah Keng Wong[1]

[1] Department of Immunology, School of Medical Sciences, Universiti Sains Malaysia, Kubang Kerian, Kelantan, Malaysia
[2] Department of Otorhinolaryngology, School of Medical Sciences, Universiti Sains Malaysia, Kubang Kerian, Kelantan, Malaysia

## ABSTRACT

Allergic rhinitis (AR) is a common disease affecting 400 million of the population worldwide. Nasal epithelial cells form a barrier against the invasion of environmental pathogens. These nasal epithelial cells are connected together by tight junction (TJ) proteins including zonula occludens-1 (ZO-1), ZO-2 and ZO-3. Impairment of ZO proteins are observed in AR patients whereby dysfunction of ZOs allows allergens to pass the nasal passage into the subepithelium causing AR development. In this review, we discuss ZO proteins and their impairment leading to AR, regulation of their expression by Th1 cytokines (i.e., IL-2, TNF-$\alpha$ and IFN-$\gamma$), Th2 cytokines (i.e., IL-4 and IL-13) and histone deacetylases (i.e., HDAC1 and HDAC2). These findings are pivotal for future development of targeted therapies by restoring ZO protein expression and improving nasal epithelial barrier integrity in AR patients.

## INTRODUCTION

Tight junction (TJ) proteins are required to form the nasal epithelial barrier and maintain its integrity. Breakdown of TJ function or expression deregulation is associated with derailed nasal epithelial barrier, leading to infiltration by allergens and subsequent development of allergic rhinitis (AR) (*Fukuoka & Yoshimoto, 2018; Steelant et al., 2016*). Moreover, growing evidence has implicated regulation of the nasal epithelial barrier integrity by histone deacetylases (HDACs), Th1 and Th2 cytokines in AR. Thus, an overall assessment and compilation of this accumulating evidence is desirable. In this review, we present and discuss the mechanisms leading to breakdown of TJs specifically on zonula occludens (ZOs), a group of important TJ proteins, as well as regulation of their expression by HDACs, Th1 and Th2 cytokines that would be informative for clinicians and researchers alike in this field.

## SURVEY METHODOLOGY

This review focuses on ZOs and their regulators i.e., HDACs, Th1 and Th2 cytokines in AR research. All articles were searched and screened by two investigators (COSS, KKW) using

Corresponding author
Kah Keng Wong, kahkeng@usm.my

the electronic databases PubMed and Google Scholar. References described in this review were obtained from the databases up to year 2019. The following keywords were used: "allergic rhinitis", "AR", "nasal epithelial barrier integrity", "zonula occludens", "ZO", "histone deacetylases", "HDACs", "Th1" and "Th2".

## Allergic rhinitis (AR)

Allergy is a hypersensitivity reaction that occurs when an individual is sensitized by allergens such as grass, tree pollen, house dust mites (HDMs), foods, insect venoms or medicines (*Azid et al., 2019*; *Sani et al., 2019*; *Tanno et al., 2016*). AR is a global health issue affecting approximately 10–25% of the population worldwide (*Elango, 2005*). AR can be characterized by events of sneezing, rhinorrhea, nasal obstruction, nasal itching and postnasal drip. It is also associated with itching of the eyes, ears and throat (*Elango, 2005*; *Pang et al., 2017*).

Onset of AR consists of two phases of reaction where the first phase involves allergen infiltration that induces the production of immunoglobulin E (IgE) and triggers the humoral immune response mediated by mast cells. The second phase is a clinical phase where the patients present with symptoms of AR as a response to subsequent antigen exposure. This involves the release of mediators such as multiple cytokines and chemokines. Nasal symptoms can be observed within minutes due to the release of neuroactive and vasoactive agents including histamine, cysteinyl leukotrienes and prostaglandin $D_2$ (*Wheatley & Togias, 2015*). The mucosa is rendered more reactive to allergens and nasal symptoms can persist for days after exposure to allergens (*Sarin et al., 2006*; *Wheatley & Togias, 2015*).

AR is also defined immunologically as an IgE-mediated inflammation reaction in the nasal airways. This is primarily due to exposure to environmental pathogens, allergens or any foreign agents that induce an inflammation reaction (*Bayrak Degirmenci et al., 2018*). These allergens contain proteases that contribute to the disruption of the airway epithelial barrier (*Runswick et al., 2007*; *Schleimer & Berdnikovs, 2017*; *Wan et al., 1999*). The interaction between IgE and dendritic cells (DCs) increases allergen uptake and its subsequent processing and presentation to naive T cells (*Sin & Togias, 2011*). Hence, higher allergen infiltration into the nasal airway increases the production of IgE in the blood. Perennial AR patients present with higher total IgE levels (*Lee et al., 2016*; *Shirasaki et al., 2011*).

## Nasal epithelial barrier integrity in AR

The nasal epithelial barrier plays an important role in sealing the nasal passage and underlying tissues from foreign pathogens by connecting the epithelial cells to each other (*London & Ramanathan Jr, 2017*; *Steelant et al., 2016*). Any intrusion from foreign particles can stimulate the production of antimicrobial host defence molecules, pro-inflammatory cytokines and chemokines by nasal epithelial cells through the activation of recognition receptors. In addition, T cells are also recruited to epithelial cells to enhance adaptive immunity.

Dysfunction of these TJ barriers can increase exposure of nasal tissues to environmental antigens. It can lead to the infusion of inflammatory cells into the lumen which contributes

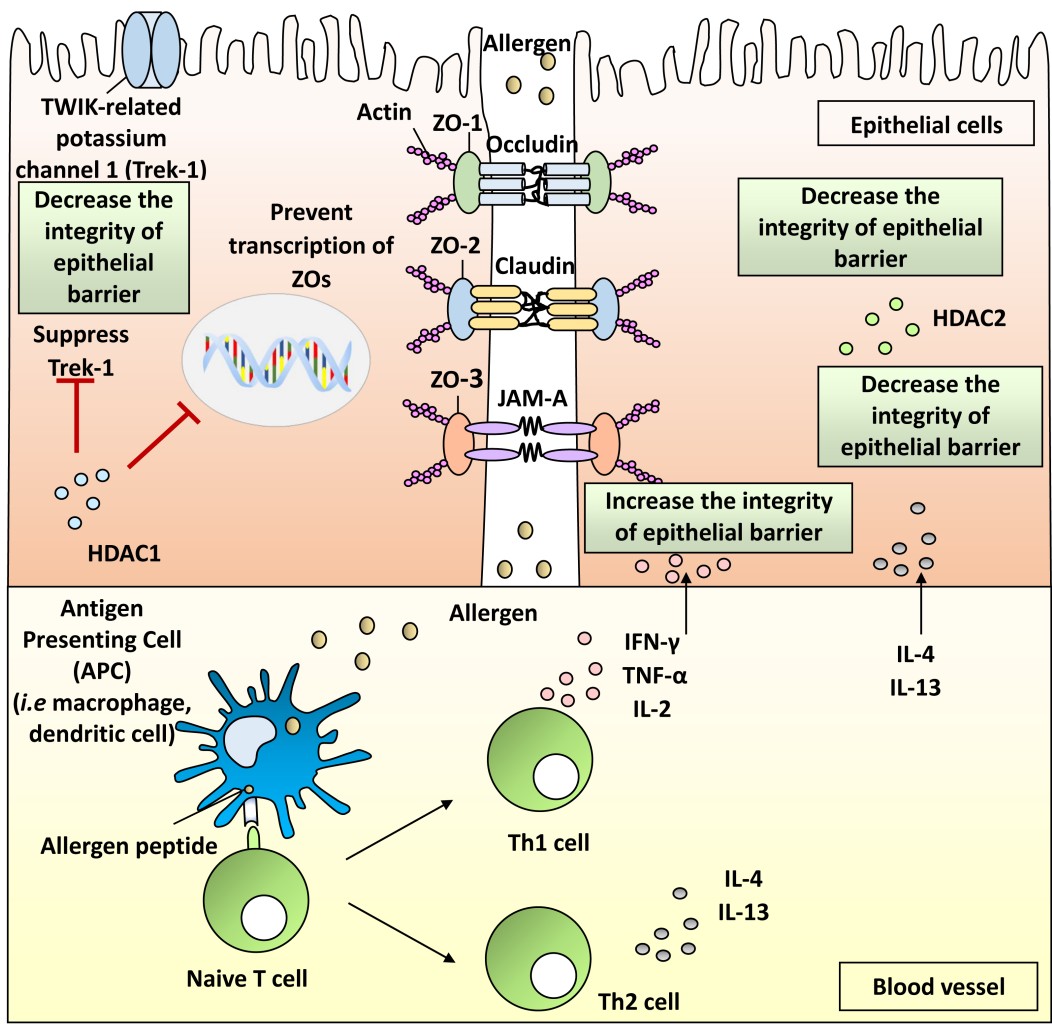

**Figure 1** Pathophysiology of allergic rhinitis (AR) from the disruption of nasal epithelial barrier and the involvement of HDACs, Th1 and Th2 cytokines.

to tissue damage or inflammation (*Soyka et al., 2012*). The disruption of the mucosal epithelial barrier has also been observed in AR animal models (*Zhang et al., 2016*).

The nasal epithelial barrier is primarily formed by cell-to-cell TJs which consist of integral membrane proteins such as claudins, occludin, junctional adhesion molecules (JAMs), as well as scaffold adaptor proteins consisting of ZO-1, ZO-2 and ZO-3 (*Beutel et al., 2019*; *London & Ramanathan Jr, 2017*). These proteins form the intercellular connection between the cells that regulate the passage of foreign pathogens (*Steelant et al., 2016*). These proteins connect together to form a complex structure that protects the epithelial barrier from inhaled pathogens (Fig. 1).

## Zonula occludens (ZO) proteins

ZO proteins are a group of key proteins associated with TJ molecules that connect transmembrane proteins to the actin cytoskeleton (*Steelant et al., 2016*). ZO proteins form an anchor directly to the underlying cytoskeleton with other TJ proteins including occludin, claudin, JAMs and tricellulin (*Bauer et al., 2010*; *Furuse et al., 1994*). ZO proteins belong to the family of membrane-associated guanylate kinase (MAGUK)-like proteins. MAGUKs are scaffolding proteins that form and maintain multimolecular complexes at distinct subcellular sites such as the cytoplasmic surface of the plasma membrane (*Bauer et al., 2010*).

ZO-1, ZO-2 and ZO-3 form a belt-like region at the outer end of intercellular space between the epithelial cells that separates the apical from the lateral plasma membrane. The proteins also play vital roles in regulating the passage of ions and molecules through the membrane (*Gonzalez-Mariscal, Betanzos & Avila-Flores, 2000*). ZO proteins consist of a multidomain structure including SRC homology 3 (SH3), guanylate kinase-like (GUK) and multiple PDZ domains (*Anderson, 1996*).

ZO-1 and ZO-2 have been detected in human nasal mucosa where ZO-1 is found in the uppermost layer of epithelium (*Kojima et al., 2013*). ZO-1 protein contains an N-terminal PDZ domain that can recognize specific C-terminal or other peptide motifs to assemble with other TJ molecules such as claudins to form a TJ barrier at gaps between epithelial cells (*Heinemann & Schuetz, 2019*; *Herve et al., 2014*; *Umeda et al., 2006*). The TJ barrier controls the diffusion of molecules by acting as semipermeable diffusion barriers through the paracellular pathway. It has been reported that transmembrane proteins such as claudin and occludin are essential for the regulation of paracellular permeability (*Balda & Matter, 2000*; *Lee, 2015*; *Roehlen et al., 2020*). ZO-1 is also responsible for the regulation of paracellular permeability (i.e., permeability for the passage of molecules between adjacent epithelial cells) via TJ complexes as it binds directly to transmembrane proteins (*Balda & Matter, 2000*; *Lee, 2015*; *Roehlen et al., 2020*). Loss of ZO-1 can retard the formation of the TJ complexes, and further breakdown of ZO-1 may result in severe disruption of the paracellular barrier in epithelial cells (*Roehlen et al., 2020*). Hence, ZO-1 plays important roles in maintaining the epithelial barrier by connecting TJ molecules to seal the epithelial cells from infiltration of environmental allergens.

## Disruption of ZO proteins in AR

The disruption of ZO proteins affects the interaction of TJ molecules, allowing the passage of allergens into the host. Decreased expression of ZO-1 in AR patients has been reported by gene expression studies (*Lee et al., 2016*; *London & Ramanathan Jr, 2017*). A study by Steelant and colleagues showed decreased levels of ZO-1 through immunofluorescent staining on AR biopsy specimens (*Steelant et al., 2016*). Furthermore, nasal epithelial cells isolated from inferior turbinate of HDM-induced AR patients demonstrated reduced *ZO-1* mRNA expression (*Steelant et al., 2018*). Likewise, the expression of ZO-1 in asthma and chronic rhinosinusitis patients was also decreased compared with healthy controls (*de Boer et al., 2008*; *Soyka et al., 2012*).

Immunofluorescence analysis of RPMI 2650, a human nasal epithelial cell line, showed a decreased of ZO-1 expression after being exposed to diesel exhaust particles (*Fukuoka et al., 2016*). Transepithelial electric resistance (TER) measurement, a procedure that assessed the integrity of TJ in cell culture of epithelial monolayers, of the RPMI 2650 was reduced in the study, and the decreased ZO-1 expression was associated with severity of AR (*Fukuoka et al., 2016*). Moreover, HDM cysteine proteinase antigen from *Dermatophagoides pteronyssinus* caused the mislocalization of ZO-1 from TJ (*Wan et al., 1999*). Hence, patients with AR demonstrate lower integrity of nasal epithelial barrier that is associated with decreased expression or disruption of ZO-1 protein.

Accumulating evidence has shown that reduced expression of ZO-1 or ZO-2 occurs in patients with chronic rhinosinusitis (CRS) without nasal polyps (*Soyka et al., 2012*) or eosinophilic esophagitis (EoE) (*Katzka et al., 2014*), respectively. CRS is characterized by mucosal inflammation involving both the nasal cavity and paranasal sinuses (*Soyka et al., 2012*), while EoE represents inflammation of the oesophagus when food antigens interact with oesophageal mucosa (*Katzka et al., 2014*). Both CRS and EoE are caused by the penetration of antigens through the gap between nasal epithelial cells (*Katzka et al., 2014*; *Soyka et al., 2012*). The expression of ZOs in these allergic diseases in both patients and animal models are summarized in Table 1.

## Histone deacetylases (HDACs) in AR

HDACs are enzymes responsible for removing acetyl group from lysine residues of target proteins. HDACs prevent gene transcription by allowing DNA to be wrapped by histones (*Jiang et al., 2015*). HDACs also promote the condensation of chromation (*Shakespear et al., 2011*). HDACs have been implicated in several inflammatory and allergic conditions including AR (*Barnes, 2013*; *Sweet et al., 2012*; *Vendetti & Rudin, 2013*). Upregulation of HDAC activity occurs in nasal epithelial cells of AR patients (*Steelant et al., 2019*).

It has been shown that expression of TJs can be increased by inhibiting the activity of HDAC1 and simultaneously decreasing the defect of epithelial barriers (*Wawrzyniak et al., 2017*). In animal models, HDAC1 protein levels in rats AR model were higher than naive rats (*Jiang et al., 2015*). Immunohistochemical results also demonstrated higher expression of HDAC1 protein in nasal epithelium of patients with sinusitis and nasal polyps contributing to the disruption of TJs (*Kaneko et al., 2017*). Furthermore, HDAC1 could supress the activity of TWIK-related potassium channel-1 (Trek-1), and Trek-1 is pivotal in the maintenance of epithelial cell barrier function (*Bittner et al., 2013*). Higher mRNA expression of *HDAC1* together with lower mRNA expression of *Trek-1* was found in nasal epithelial cells from patients with AR compared with healthy subjects (*Wang et al., 2015*).

ZO-1 expression was previously shown to be decreased in the presence of HDAC1. Lower levels of *ZO-1* mRNA expression were observed in AML-12 murine hepatocyte cells that overexpressed HDAC1 (*Lei et al., 2010*). Studies on epithelial-mesenchymal transition (EMT), an oncogenic process that induces epithelial cells to transform into anchorage-independent mesenchyme-like cells for increased metastatic capabilities of cancer cells, also showed an association with HDAC1 and ZO-1 (*Zhou et al., 2015*). ZO-1 is involved in

Siti Sarah et al. (2020), *PeerJ*, DOI 10.7717/peerj.9834

**Table 1  The expression of zonula occludens (ZOs) in human and animal models.**

| ZOs | Samples | Treatment | Change in expression | References |
|---|---|---|---|---|
| ZO-1 | **Treatment with Th1 cytokines:** | | | |
| | Nasal mucosa from normal wild type BALB/c mice | TNF-$\alpha$, IFN-$\gamma$ | Downregulated | *Steelant et al. (2018)* |
| | Nasal epithelial cells from HDM-induced AR patients | IFN-$\gamma$ | Downregulated | *Steelant et al. (2016)* |
| | **Treatment with Th2 cytokines:** | | | |
| | Nasal mucosa from normal wild type BALB/c mice | IL-4 | Downregulated | *Steelant et al. (2018)* |
| | Nasal epithelial cells from HDM-induced AR patients | IL-4 | Downregulated | *Steelant et al. (2016)* |
| | Calu-3 cells (human epithelial nasal cell lines) | IL-4 and IL-13 | Downregulated | *Fukuoka & Yoshimoto (2018)* |
| | **Other treatment:** | | | |
| | Human nasal epithelial cell line, RPMI 2650 | Cultured with diesel exhaust particle (DEP) | Downregulated | *Fukuoka et al. (2016)* |
| | Nasal epithelial cells in AR mice | Intranasal administration with DEP | Downregulated | *Fukuoka et al. (2016)* |
| | Nasal biopsy specimens from HDM-induced AR patients | No treatment | Downregulated | *Steelant et al. (2016)* |
| | Bronchial epithelium biopsy from asthmatic patients | No treatment | Downregulated | *de Boer et al. (2008)* |
| | Nasal epithelial cells from septal surgery patients | Treated with *Alternaria alternate* | Downregulated | *Shin et al. (2019)* |
| ZO-2 | Nasal biopsy specimens from chronic rhinosinusitis patients | No treatment | Downregulated | *Soyka et al. (2012)* |
| ZO-3 | Esophageal epithelia biopsy samples from patients with eosinophilic esophagitis (EoE) | Treated topical fluticasone | Upregulated | *Katzka et al. (2014)* |
| | Esophageal epithelia biopsy samples from patients with EoE | No treatment | Downregulated | *Katzka et al. (2014)* |

EMT where loss of ZO-1 expression can induce invasion of cancer cells. Higher HDAC1 mRNA and protein expression levels were found in hepatocellular carcinoma (HCC) cell lines (HepG2, Hep3B, Huh7, PLC/PRF/5, SK-Hep-1) compared with normal human epithelial cell line (THLE-3) (*Zhou et al., 2015*). Inhibition of HDAC1 in these HCC cells showed an increase of ZO-1 mRNA and protein expression, leading to decreased invasion capabilities of HCC cells (*Zhou et al., 2015*). Thus, ZO-1 expression can be inhibited by HDAC1 leading to breakdown of epithelial cells' anchorage, and it remains unknown if similar effects might also occur in nasal epithelial cells.

In contrast with HDAC1, evidence has shown that HDAC2 expression is required to prevent breakdown of nasal epithelial barrier integrity in AR. Decreased levels of HDAC2 were observed in patients with asthma and asthmatic smoking patients, as in patients with chronic obstructive pulmonary disease (*Bhavsar, Ahmad & Adcock, 2008*). Higher levels of HDAC2 can restore steroid sensitivity in asthmatic patients (*Bhavsar, Ahmad & Adcock, 2008*), and nasal scrape samples of patients with persistent AR showed weak expression of HDAC2 (*Sankaran et al., 2014*). Moreover, deficiency of HDAC2 in intestinal epithelial cells (IEC) of mice was associated with chronic basal inflammation (*Turgeon et al., 2013*). Deletion of HDAC2 from IEC displayed an increased permeability to fluorescein isothiocyanate-dextran 4kDa (FD4; a fluorochrome for investigation of cell permeability) by assessing the intensity of fluorescence in the mice blood (*Turgeon et al., 2013*), and increased penetration by FD4 indicated increased leakiness that may be due to disruption of epithelial barrier.

However, downregulation of HDAC2 with the treatment of Trichostation-A (TSA), an HDAC inhibitor (HDACi), increased the expression of *ZO-1* mRNA in fetal human lens epithelial cells (*Ganatra et al., 2018*). TSA treatment in this study decreased the association between HDAC2 with the promoter region of *ZO-1* as demonstrated by chromatin immunoprecipitation assay (*Ganatra et al., 2018*). The effect of HDAC2 inhibitor CAY10683 was investigated on the expression on ZO-1 at the intestinal mucosal barrier of lipopolysaccharide (LPS)-stimulated NCM460 cells (a normal human colon mucosal epithelial cell line) (*Wang et al., 2018*). LPS was used to induce damage to the mucosal barrier of NCM460 cells. The NCM460 cells treated with the HDAC2 inhibitor (CAY10683) increased mRNA and protein levels of ZO-1 (*Wang et al., 2018*). Collectively, this suggests that HDAC2 plays differential roles in the increase or reduction of epithelial barrier integrity depending on the site of the human epithelial cells. HDAC2 prevents the breakdown of nasal epithelial barrier but it may promote the opposite effect in human lens or colon mucosal epithelial cells via downregulation of ZO-1 expression.

Inhibiting HDAC activities with HDACi (JNJ-26481585) may be able to restore the structure of ZO molecules in nasal epithelial cells (*Steelant et al., 2019*). In the same study, immunofluorescent staining showed that ZO-1 expression was significantly weaker in AR patients compared with healthy controls, and further treatment with JNJ-26481585 increased the expression of ZO-1 protein.

The HDACi sodium butyrate (SoB) is a short chain fatty-acid produced by the microbial fermentation of dietary fibre in colonic lumen (*Bordin et al., 2004*). The Rat-1 fibroblasts cell line expresses ZO-1 and ZO-2 proteins (*Bordin et al., 2004*). When the cells lysates were

cultured in the presence of SoB, densitometric analysis of immunoblots showed that ZO-1 and ZO-2 levels were upregulated (*Bordin et al., 2004*). Collectively, HDAC1 and HDAC2 suppress the expression of ZO proteins leading to breakdown of epithelial cells barrier integrity as demonstrated by these studies either in AR or non-AR epithelial cells.

## Th1 cytokines in AR

Cytokines play an important role in mediating allergic inflammation. The roles of Th2 cytokines in AR have been well-documented (*Steelant et al., 2016*; *Sun et al., 2020*; *Zhao et al., 2017*). Imbalance of Th1 and Th2 cytokines appears to be involved in the AR inflammatory pathway (*Zhao et al., 2017*). However, there is a lack of review on Th1 cytokines and their roles in the breakdown of nasal epithelial barrier integrity. Moreover, dysfunctional Th1 responses have been proposed to be responsible for the exaggerated Th2 responses that occur in AR patients (*Eifan & Durham, 2016*). Th1 cells produce IL-2, IFN-$\gamma$ and TNF-$\alpha$ in response to allergic inflammation (*Ackaert et al., 2014*). Th1 cytokines can cause disruption of TJ molecules including ZO proteins in nasal epithelial barrier, leading to allergic inflammation.

Th1 response is characterized by IFN-$\gamma$ production which stimulates bactericidal activities of macrophages and boosts immunity against intracellular pathogens and virus infection (*Marshall et al., 2018*). IFN-$\gamma$ plays a key role in bridging the innate and adaptive immune systems (*Bayrak Degirmenci et al., 2018*). It is also essential in the regulation of local leukocyte-endothelial interaction (*Akkoc et al., 2008*).

IFN-$\gamma$ increases the permeability of primary bronchial epithelial cells and T84 colonic epithelial cells by disassembling TJ structures (*Bruewer et al., 2005*). In order to observe the expression of *ZO-2* in CRS patients, human epithelial cells were treated on air-liquid interface (ALI) culture with IFN-$\gamma$. The results showed that opening of TJs between the neighbouring cells occurred in patients compared with healthy controls (*Soyka et al., 2012*). However, no significant decrease of ZO-1 expression in AR patients was observed when the epithelial cells were treated with IFN-$\gamma$ and TNF-$\alpha$ cytokines (*Lee et al., 2016*). Additionally, cultured primary nasal epithelial cells in ALI stimulated with TNF-$\alpha$ and IFN-$\gamma$ showed a decrease of epithelial barrier integrity *in vitro* (*Steelant et al., 2018*).

Furthermore, expression of ZO-1 protein in primary airway cells from cystic fibrosis patients was reduced in the presence of IFN-$\gamma$ and TNF-$\alpha$ cytokines (*Coyne et al., 2002*). Prolonged exposure of IFN-$\gamma$ and TNF-$\alpha$ to the cell culture led to a significant damage to ZO-1 molecules (*Coyne et al., 2002*). This damage caused an increase of cell permeability to external solutes and a decrease in transepithelial resistance. Further investigation of wild type BALB/c mice endonasally instilled with IFN-$\gamma$ and TNF-$\alpha$ increased the FD4 mucosal barrier permeability associated with decreased ZO-1 expression *in vivo* (*Steelant et al., 2018*).

However, in AR mice model and AR patients, Th1 cytokines have been associated with increased expression of TJ molecules and decreased AR severity, respectively. Lower levels of Th1 cytokines, IL-2 and IFN-$\gamma$ were detected in the serum sample from OVA-sensitized mice with AR compared with controls (*Wang et al., 2016*). When the OVA-sensitized mice were treated with SoB, IL-2 and IFN-$\gamma$ levels were increased, leading to increased

expression of TJ molecules (*Wang et al., 2016*). The levels of IFN-$\gamma$ in plasma sample of AR patients was significantly lower compared with healthy controls (*Bayrak Degirmenci et al., 2018*). The same study showed that downregulated levels of Th1 cytokines were associated with higher severity of AR symptoms. Furthermore, the levels of IFN-$\gamma$ were inversely correlated with higher nasal symptoms scores as measured by evaluating the severity of sneezing, nasal itching, nasal obstruction and watery nasal discharge (*Bayrak Degirmenci et al., 2018*). Further mechanistic studies are recommended to elucidate whether Th1 cytokines exert their protective effects on nasal epithelial barrier integrity via increased TJ molecules expression in human AR cells.

## Th2 cytokines in AR

The involvement of Th2 cytokines in AR has been widely investigated. The serum levels of Th2 cytokines including IL-4 and IL-13 are elevated in AR patients (*Jordakieva & Jensen-Jarolim, 2018*). Increased expression of IL-4 in nasal epithelial cells of HDM-induced AR patients reduced *ZO-1* mRNA expression (*Steelant et al., 2016*). Breakdown of the epithelial barrier was observed after stimulation of nasal epithelial cells with IL-4 and significantly increased the permeability of FD4 (*Steelant et al., 2016*).

Both IL-4 and IL-13 play critical roles in promoting B cells to produce IgE (*Shirkani et al., 2019*; *Zhao et al., 2017*). Protein levels of IL-4 and IL-13 in nasal mucosa of guinea pig of AR-sensitized pig were higher compared with controls (*Zhao et al., 2017*). This was supported by findings where higher serum levels of IL-4 and IL-13 were found in AR-sensitized pigs compared with controls (*Zhao et al., 2017*). In addition, treatment of lung cancer cells (Calu-3) with *IL-4* and *IL-13* reduced the protein expression of ZO-1 protein (*Fukuoka & Yoshimoto, 2018*).

Immunofluorescent staining of human bronchial epithelial cells of asthmatic patients demonstrated that disruption of TJs in the ALI cultures occurred and weak expression of ZO-1 was observed (*Wawrzyniak et al., 2017*). Blocking IL-4 and IL-13 in asthma patients did not show difference in TER measurement (*Srinivasan et al., 2015*; *Wawrzyniak et al., 2017*). However, nullifying the effects of IL-4 and IL-13 using anti-IL4 and anti-IL-13 supplemented to the ALI culture of control bronchial epithelial cells *in vitro* enhanced the TER measurement (*Wawrzyniak et al., 2017*). Moreover, *IL-4* and *IL-13* mRNA expression levels were increased together with downregulated *ZO-1* mRNA expression in the jejunum of OVA-sensitized rats (*Tulyeu et al., 2019*).

Downregulation of *ZO-1* mRNA expression potentially through regulation by Th2 cytokine was also observed *in vivo*. Endonasal stimulation of wild-type BALB/c mice with IL-4 and IL-13 demonstrated increased FD4 permeability associated with reduced *ZO-1* mRNA expression compared with saline-instilled mice (*Steelant et al., 2018*). Taken together, these studies indicate that IL-4 and IL-13 contribute to the breakdown of nasal epithelial barrier by reducing the expression of ZO-1.

## CONCLUSION

In conclusion, HDAC1 and HDAC2 play pathogenic roles in the breakdown of nasal epithelial barrier integrity via suppression of ZO proteins expression. This is potentially

regulated by Th2 cytokine signaling pathways as higher levels of Th2 cytokines in AR patients are accompanied with decreased epithelial barrier integrity and ZO-1 expression. In contrast, higher levels of Th1 cytokines appear to preserve the nasal epithelial barrier integrity of AR patients. Future research should investigate and compare which specific HDACi or blocking antibodies of Th2 cytokines demonstrate potent restoration of ZO proteins expression in nasal epithelial cells of AR animal models, as well as ameliorating their symptoms. Targeting these pathogenic pathways might be effective in AR therapy to maintain the expression and structure of ZOs at the nasal epithelial barrier.

### Funding

This work was supported by grants from Universiti Sains Malaysia comprising of the Bridging Grant (304.PPSP.6316332) awarded to Kah Keng Wong and Research University Grant (1001.PPSP.8012285) awarded to Mohd Ashari Noor Suryani. The funders had no role in study design, data collection and analysis, decision to publish, or preparation of the manuscript.

### Grant Disclosures

The following grant information was disclosed by the authors:
Universiti Sains Malaysia comprising of the Bridging Grant: 304.PPSP.6316332.
Research University Grant: 1001.PPSP.8012285.

### Competing Interests

The authors declare there are no competing interests.

### Author Contributions

- Che Othman Siti Sarah conceived and designed the manuscript, prepared the figure and table, wrote the manuscript, and approved the final draft.
- Norasnieda Md Shukri and Noor Suryani Mohd Ashari reviewed drafts of the paper, and approved the final draft.
- Kah Keng Wong conceived and designed the manuscript, wrote parts of the manuscript, reviewed drafts of the paper, and approved the final draft.

### Data Availability

This is a literature review; there is no raw data or code.

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
