# Peer review of "Zonula occludens and nasal epithelial barrier integrity in allergic rhinitis"

_PeerJ, doi:10.7717/peerj.9834_

## Round 0.1 · original submission · Major Revisions

Two expert reviewers have read your manuscript and provided feedback to me. While both agree that there is potential in this review article, they have pointed out that it needs major improvement in terms of the clarity of the writing and the transitions between sections, and that it would greatly benefit from editing by a native English speaker before being ready for publication. If you choose to submit a revised manuscript, please address these major concerns as well as all the points raised by reviewer #1 in his comments below and in the annotated pdf file that he has attached.

Reviewer 1 ·

Basic reporting

The review has good references, is cross disciplinary and introduces the subject clearly. However, the field has been reviewed recently:

https://www.sciencedirect.com/science/article/pii/S1323893017301594?via%3Dihub

https://www.ncbi.nlm.nih.gov/pmc/articles/PMC5752131/

However, this review focuses more on HDAC inhibitors that the other reviews.

Experimental design

The review has good organization, but it needs to be edited to flow better. Presently all of the data is there, but it is more of a list of study findings organized by topic.

Validity of the findings

The conclusions are well stated, but introductory sentences are not always present and more work is needed to identify unresolved questions and future directions. Also the concept that ZO-1 is a scaffolding protein for the tight junction and how it may regulate the paracellular pathway should be brought up in more detail. For example, how might it translate to barrier function?

Additional comments

I have attached some minor comments in the pdf as well

Annotated reviews are not available for download in order to protect the identity of reviewers who chose to remain anonymous.

---

## Round 0.2 · Minor Revisions

Thank you for addressing the concerns and making the edits that the Reviewer requested. I have read over the revised version carefully. There were a number of small grammar issues, so I took the liberty of fixing these in the attached pdf document.

I also include several comments that need to be addressed before this can be accepted for publication. Specifically, there are several sections that include contradictory statements, presumably because the literature has conflicting results. But the way it is presented is extremely confusing, since you will state an overall topic sentence, followed by some supporting studies, and then include some completely contradictory evidence without any discussion or explanation. You need to provide some context or explanation for such contradictory findings, either taken from the primary literature or from your own personal perspective.

---

## Round 0.3 · accepted · Accept

Thank you for addressing the concerns in this revised version. I am happy to accept this review article for publication. Congratulations!